# Apolipoprotein E (ApoE) Rescues the Contractile Smooth Muscle Cell Phenotype in Popliteal Artery Aneurysm Disease

**DOI:** 10.3390/biom13071074

**Published:** 2023-07-04

**Authors:** Jessica Pauli, Tessa Reisenauer, Greg Winski, Nadja Sachs, Ekaterina Chernogubova, Hannah Freytag, Christoph Otto, Christian Reeps, Hans-Henning Eckstein, Claus-Jürgen Scholz, Lars Maegdefessel, Albert Busch

**Affiliations:** 1Department for Vascular and Endovascular Surgery, Klinikum rechts der Isar, Technical University Munich, 81675 Munich, Germany; 2German Center for Cardiovascular Research (DZHK), Partner Site Munich Heart Alliance, 10785 Berlin, Germany; 3Molecular Vascular Medicine Group, Center for Molecular Medicine, Karolinska Institute, 17177 Stockholm, Sweden; 4Perioperative Medicine and Intensive Care, Karolinska University Hospital, 17177 Stockholm, Sweden; 5Department of General, Visceral, Transplantation, Vascular & Pediatric Surgery, University Hospital Würzburg, 97080 Würzburg, Germany; 6Division of Vascular and Endovascular Surgery, Department for Visceral, Thoracic and Vascular Surgery, Medical Faculty Carl Gustav Carus and University Hospital, Technische Universität Dresden, 01307 Dresden, Germany; 7Wisplinghoff Laboratories, 50858 Cologne, Germany

**Keywords:** popliteal artery aneurysm, apolipoprotein E, vascular smooth muscle cell, phenotype switch, proliferation

## Abstract

Popliteal artery aneurysm (PAA) is the most frequent peripheral aneurysm, primarily seen in male smokers with a prevalence below 1%. This exploratory study aims to shed light on cellular mechanisms involved in PAA progression. Sixteen human PAA and eight non-aneurysmatic popliteal artery samples, partially from the same patients, were analyzed by immunohistochemistry, fluorescence imaging, Affymetrix mRNA expression profiling, qPCR and OLink proteomics, and compared to atherosclerotic (*n* = 6) and abdominal aortic aneurysm (AAA) tissue (*n* = 19). Additionally, primary cell culture of PAA-derived vascular smooth muscle cells (VSMC) was established for modulation and growth analysis. Compared to non-aneurysmatic popliteal arteries, VSMCs lose the contractile phenotype and the cell proliferation rate increases significantly in PAA. Array analysis identified APOE higher expressed in PAA samples, co-localizing with VSMCs. APOE stimulation of primary human PAA VSMCs significantly reduced cell proliferation. Accordingly, contractile VSMC markers were significantly upregulated. A single case of osseous mechanically induced PAA with a non-diseased VSMC profile emphasizes these findings. Carefully concluded, PAA pathogenesis shows similar features to AAA, yet the mechanisms involved might differ. APOE is specifically higher expressed in PAA tissue and could be involved in VSMC phenotype rescue.

## 1. Introduction

Aneurysm formation is most often seen and has been best investigated for the ascending aorta and specifically the infrarenal abdominal aortic aneurysm (AAA). The most frequent peripheral aneurysm and a common concern in vascular surgery is the popliteal artery aneurysm (PAA). Its prevalence is about 1% in the susceptible population (male smokers, aged > 55) and occurs bilaterally upon diagnosis in more than 50% of cases [1]. Approximately half of the affected patients show at least one further aneurysm, most often in the aorta [2,3].

Despite a similar population, clinical symptoms are distinct. The rupture rate is low and complications are mainly caused by leg ischemia, due to silent onset longtime or acute peripheral embolism or local compression resulting in acute leg ischemia, deep vein thrombosis or popliteal fossa pain [4]. This has led to a vivid ongoing discussion about treatment indication and modality. Currently, open repair by local aneurysm exclusion or bypass is favored over endovascular covered stent implantation [5,6]. Generally, indications for elective repair are a maximum diameter >20 mm, luminal thrombus load and possible eccentric morphology [4,7,8,9]. Occurrence is occasionally within defined clinical syndromes (e.g., Loeys-Dietz syndrome) or rheumatoid vasculitis (e.g., Behcet’s disease) or exostoses, causing eventual mechanic vessel irrigation [10,11,12,13]. In most cases, PAA should be considered a distinct entity with a widely unknown pathogenesis [9].

This might be due to the specific anatomic niche with constant movement in at least two axis and the muscular type artery origin with a possible different susceptibility to external stimuli in comparison to elastic type arteries, such as the aorta [14]. Nevertheless, PAA samples show features of degenerative vascular wall remodeling [15,16]. This was mainly investigated compared to characteristics of AAA and is equally lacking hints of the initial cause for diameter enlargement [17]. Here, in the so-called VSMC phenotype switch, the loss of contractile characteristics is a pathogenic hallmark and phenotype rescue is a proposed salvage mechanism to halt AAA growth [18,19,20]. Furthermore, trans-differentiation to other cell types has been described, once the senescent phenotype is lost [19,21].

Specifically for PAA, neutrophil and T_H1_-cell rich infiltrates and upregulation of proteolytic matrix metalloproteinases (MMP), and cathepsins have been observed in tissue samples [22,23]. Additionally, loss of contractile elements and macrophage infiltration has been described by others and us [16,23,24]. Apart from that, little is known about specific pathomechanisms involved in PAA development and progress, yet severe atherosclerosis is concomitantly observed in most patients. In the present study, we aimed to investigate specific features of PAA in comparison to non-aneurysmatic popliteal arteries based on histologic examination, RNA profiling and cell culture experiments to identify and modify potential mechanisms distinct to AAA or atherosclerosis.

## 2. Materials and Methods

Tissue acquisition: Tissue acquisition was in accordance with the declaration of Helsinki, with approval of the local ethic review committee (University of Würzburg 20181107_02; Technical University Munich: 2799/10) and with patients informed and written consent. If possible, PAAs were collected with an adjacent non-aneurysmatic vessel via a medial surgical approach.

All patients were operated on due to acute or critical limb ischemia for peripheral arterial occlusive disease (PAOD) or on preemptive means when a threshold of 20 mm diameter was reached for PAA. Tissue was immediately rinsed in PBS and divided for formalin fixation (3.5% formaldehyde, Fischer, Achern-Fautenbach, Germany) and snap freezing in liquid nitrogen. Detailed patient characteristics and numbers of samples are shown in Appendix A.

AAA and PAOD tissue samples were available for comparative purposes from the Munich Vascular Biobank as previously described in extenso [25]. Briefly, samples from intraoperative specimen were treated in the same way as PAA samples described here.

Primary cell culture: Primary human popliteal artery SMCs from patients were isolated from PAA biopsies, harvested during surgical repair and stored in complete DMEM/F12 Medium (Sigma Aldrich, Taufkirchen, Germany) containing 5% Fetal Bovine Serum (Gibco, Thermo Fisher Scientific) and 1% Pen-Strep (Gibco, Thermo Fisher Scientific, Darmstadt, Germany). The tissue was placed in a sterile petri dish and washed with PBS (Gibco, Thermo Fisher Scientific). Adventitia was removed, and the remaining media was cut into small pieces using a sterile scalpel. The pieces of tissue were placed in digestion medium (1.4 mg/mL Collagenase A, Roche, Mannheim, Germany, in complete DMEM/F12 Medium) in a humidified incubator at 37 °C and 5% CO_2_ for 4–6 h. Cells were strained using a 100 μm cell strainer to remove debris. After 2 washing steps (centrifuge 400 g, 5 min; discard supernatant, re-suspended in 15 mL complete DMEM/F12 Medium) cells were re-suspended in 7 mL complete DMEM/F12 Medium and placed in a small cell culture flask in a humidified incubator at 37 °C and 5% CO_2_. Upon confluence, cells were stored in liquid nitrogen or processed immediately.

Primary human aortic artery smooth muscle cells (hAoSMCs) were obtained from PeloBiotech (#PB-CH-280-2011) and cultured in Smooth Muscle Cell Growth Medium (PeloBiotech, Munich, Germany), following manufacturer’s instructions. These cells served as control for all cell-based assays.

All cells are (isolated and commercially available ones) are cultured in Smooth Muscle Cell Growth Medium (PeloBiotech, Munich, Germany), following manufacturer’s instructions. All cells are used until passage 7.

HE staining, light microscopy and digital image acquisition: 2-μm sections of paraffin-embedded samples were mounted on Superfrost© slides (Menzel, Gießen, Germany) and stained with hematoxylin/eosin (HE) and Elastica van Gieson according to the manufacturer’s protocol. Slides were digitalized with a NanoZoomer 2.0-HT Digital slide scanner: C9600© and pictures were taken with the NDP.view2© software (both Hamamatsu, Kyoto, Japan).

Immunohistochemistry: For immunohistochemistry (IHC), sections were mounted on 0.1% poly-L-lysine (Sigma-Aldrich, St. Louis, MO, USA) pre-coated SuperFrost Plus slides (Thermo Fisher Scientific, Waltham, MA, USA). For antigen retrieval the slides boiled in a pressure cooker with 10 nM citrate buffer (distilled water with citric acid monohydrate, pH 6.0) and endogenous peroxidase activity was blocked with 3% hydrogen peroxide. Samples were blocked with 10% goat serum and then incubated with primary antibodies diluted in Dako REAL Antibody Diluent (Dako, Glostrup, Denmark). Slides were then treated with biotinylated secondary antibodies and target staining was performed with peroxidase-conjugated streptavidin and DAB chromogen (Dako REAL EnVision Detection System Peroxidase/DAB+, Rabbit/Mouse Kit; Dako, Glostrup, Denmark). Mayer’s hematoxylin (Carl Roth, Karlsruhe, Germany) was used for counterstaining, with appropriate positive and negative controls for every target antibody. All slides were scanned with Aperio AT2 (Leica, Wetzlar, Germany) and images were taken with the Aperio ImageScope software (version: 12.3, Leica, Wetzlar, Germany).

Antibody list: ApoE 1:800 (ab52607 abcam), SMA 1:200 (M0635 Dako), CD68 1:2000 (M0814 Dako), Ki67 1:50 (ab16667 abcam), SMA 1:200 (ab5694 abcam), vimentin 1:200 (ab8978 abcam), desmin 1:100 (ab32362 abcam).

KI67-positive cell counting: For assessment of proliferation rates in the aneurysm and vessel wall, Ki67-positive cells were counted manually within three high power fields (HPF) per sample in all probes 15 vs. 8 probes included in the study and then average was calculated (HPF: 40× objective lens).

Immunofluorescence staining (tissue): For immunofluorescent stainings, 2-μm sections were mounted on 0.1% poly-L-lysine (Sigma-Aldrich, St. Louis, MO, USA) pre-coated SuperFrost Plus slides (Thermo Fisher Scientific, Waltham, MA, USA). For antigen retrieval, the slides were boiled in a pressure cooker with 10 nM citrate buffer (distilled water with citric acid monohydrate, pH 6.0) and endogenous peroxidase activity was blocked with 3% hydrogen peroxide. Two antibodies were applied after one another. For each antibody, samples were blocked for 1 h (5% horse serum, 1%BSA, 0,5% Triton-X100) and incubated with primary antibodies diluted in 5% horse serum overnight (ApoE 1:600 ab52607 and SMA 1:200 ab7817, both abcam). The appropriate secondary antibody was added for 1 h (in 5% horse serum) on the next day (Alexa488 goat anti-mouse 1:400 A11001 and Alexa647 goat anti-rabbit 1:400 A21245, both Thermo Fisher). Autofluorescence quenching and counter-staining with DAPI were performed. There were appropriate positive and negative controls for every target antibody. Images were acquired with a Zeiss Axioscan7 Scanner, using ZEN 3.3 software (Carl Zeiss Microscopy GmbH, Jena, Germany). In total, 4–6 patient samples were stained and the most representative ones were chosen for the composite figures.

Immunofluorescent staining (cells): Cells were seeded in 4-well chamber CultureSlides (REF354104, Falcon, Miami, FL, USA) and fixed with 4% PFA for 15 min. Then, permeabilization (1% saponin) and blocking (5% BSA, 0.3% Triton X-100) were performed. After that, cells were incubated with primary antibodies diluted in 3% goat serum + 0.3% Triton X-100 overnight (SM22 1:200 ab14106 and SMA 1:300 ab5691, both abcam). The appropriate secondary antibody was added for 1 h (3% goat serum + 0.3% Triton X-100) the next day (Alexa488 goat anti-rabbit 1:400 A11034, Thermo Fisher). Then, Phalloidin staining was performed: Phalloidin (1:80 A22287. Thermo Fisher) was diluted in 1% BSA + 0.3 % Triton X-100 and incubated for 30 min before counterstaining with DAPI. Appropriate negative controls were performed for every target antibody. Images were acquired with a Leica THUNDER Imager using LAS X software (Leica Microsystems, Germany) and Fiji Image J.

RNA isolation and qPCR (tissue): Tissues were cut in ~50 mg (approx. 3 × 3 × 3 mm) pieces with a scalpel on dry ice using a micro scale. Tissue was homogenized in 700 μL Qiazol lysis reagent and total RNA was isolated using the miRNeasy Mini Kit (Qiagen, Venlo, The Netherlands) according to manufacturer’s instruction. RNA concentration and purity were assessed using the NanoDrop. Next, first strand cDNA synthesis was performed using the High-Capacity-RNA-to-cDNA Kit (Applied Biosystems, Waltham, MA, USA), following the manufacturer’s instruction.

Quantitative real-time TaqMan PCR was then performed using Taqman Gene Expression Master Mix (Thermo Fisher, Darmstadt, Germany). PCR was run on a QuantStudio3 Cycler (Applied Biosystems, Waltham, MA, USA) using 96 well plates. Gene expression was normalized to Rplp0 and quantified with the 2^ΔΔCt^ method.

Primer list: APOE (Hs00171168_m1), LDLR (Hs01092524_m1), LRP1 (Hs00233856_m1), ABCA1 (Hs01059101_m1), ABCG1 (Hs00245154_m1), MMP9 (NM 004994.2), MYH11 (NM 002474.2); ACTB (NM_001101); MMP9 (NM 004994.2); MYH11 (NM 002474.2)

Affymetrix mRNA expression profile analysis: RNAs from aneurysm and control tissue samples were reverse transcribed, labeled and hybridized to Affymetrix PrimeView (Thermo Fisher, Germany) gene expression microarrays according to the manufacturer’s instructions. Data analysis was performed in R with Bioconductor packages. Briefly, data were read with package affy, normalization was performed with package vsn and differentially expressed genes were defined with package limma [26]. The raw data are shown in Appendix A.

Cell treatment, RNA isolation and qPCR (Primary PAA cells/human aortic SMCs): Cells were placed in 12-well plates (triplicates each) and treated with 50 μg/mL ApoE (ab280330, Abcam, Cambridge, UK) in OptiMEM +2% FBS (Thermo Fisher, Dreieich, Germany) or OptiMEM +2%FBS only (control-treatment) for 6 h, 24 h, 48 h and 72 h. Cells were washed with PBS and harvested with 300 μL Qiazol lysis reagent. Total RNA was isolated using the RNeasy Mini Kit (Qiagen, Venlo, The Netherlands) according to manufacturer´s instruction. RNA concentration and purity was assessed using the NanoDrop. Next, first strand cDNA synthesis was performed using the High-Capacity-RNA-to-cDNA Kit (Applied Biosystems, USA), following the manufacturer’s instruction.

Quantitative real-time TaqMan PCR was then performed using Taqman Gene Expression Master Mix (Thermo Fisher, Germany). PCR was run on a QuantStudio3 Cycler (Applied Biosystems, USA) using 96-well plates. Gene expression was normalized to Rplp0 and quantified with the 2^ΔΔCt^ method.

Primer list: RPLP0 (Hs00420895_gH), MYOCD (Hs00538076_m1), TAGLN (Hs01038777_g1), ACTA2 (Hs00426835_G1), SMTN (Hs01022255_g1), CNN1 (Hs00959434_m1), KLF4 (Hs00358836_M1), CD34 (Hs02576480_m1), CD44 (Hs01075864_m1), VIM (Hs00958111_m1), COL1A1 (Hs00164004_M1).

WESTERN BLOT: Cells were placed in 6-well plates (triplicates each) and treated with 50 μg/mL ApoE in OptiMEM (Thermo Fisher, USA, Lenvatinib-treatment) or OptiMEM (control-treatment) for 48 h. Cells were washed with ice-cold PBS and harvested with 100 μL freshly prepared complete RIPA Buffer (RIPA Lysis and Extraction Buffer, Thermo Fisher, USA) containing Phosphatase Inhibitor Cocktail 2 and 3 (Sigma Aldrich, St. Louis, MO, USA) and Halt™ Protease Inhibitor Cocktail (Thermo Fisher, Waltham, MA, USA). After homogenization with a pistil lysate was frozen at −80 °C in aliquots of total protein lysate.

Following the manufacturer’s instruction, total protein concentration was measured using the Pierce™ BCA Protein Assay Kit (Thermo Fisher, USA).

An amount of 10 μg of protein of each sample was denatured and reduced at 95 °C for 10 min, then separated in a Bolt™ 4–12% Bis-Tris Plus Gel (Thermo Fisher, USA) and transferred onto Trans-Blot^®^ Turbo™ Transfer Pack Membranes (BioRad, Hercules, CA, USA). The blots were blocked with 5% BSA in Tris-buffered saline + 0.1% Tween-20 for 1 h, followed by overnight incubation with the primary antibody against SMTN (ab8969, Abcam, UK), CNN1 (ab46794, Abcam, UK), and SM22 (ab14106, Abcam, UK) in TBS-T + 5% BSA. After washing with TBS-T, blots were then incubated with anti-mouse- or anti-rabbit HRP (horseradish peroxidase)-conjugated secondary antibodies (ab205718, ab205719, Abcam, UK) and visualized using ECL Prime Western Blotting Detection Reagent (Amersham, Amersham plc, UK) in the C600 Azure Biosystems Imager 585 (Biozym)/ChemiDoc XRS System (Bio-rad, Hercules, CA, USA). Blots were stripped using Restore™ Plus Western Blot Stripping Buffer (Thermo Fisher, USA), blocked again with 5% BSA in TBS-T and incubated with the primary antibody against ß-Actin (A1978-200 μL, Sigma Aldrich, USA) in TBS-T + 5% BSA for 1 h. After washing with TBS-T, blots were again incubated with anti-mouse-HRP (horseradish peroxidase)-conjugated secondary antibody (ab205719, Abcam, UK) and visualized using ECL Prime Western Blotting Detection Reagent (Amersham, Amersham plc, UK) in the C600 Azure Biosystems Imager 585 (Biozym)/ChemiDoc XRS System (BioRad, Hercules, CA, USA) Antibodies and dilutions: SMTN, ab8969: 1:1000; CNN1, ab46794: 1:5000; SM22, ab14106: 1:1000, ß-Actin, A1978: 1:8000; Goat-Anti-Rabbit IgG H&L HRP, ab205718: 1:10,000; Goat-Anti-Mouse IgG H&L HRP, ab205719: 1:10,000.

The blots were quantitated by using Fiji ImageJ Software. (Complete membranes/gels are shown in Appendix A).

Dynamic live-cell imaging assay: Dynamic live-cell imaging experiments were performed following the instructions provided by Essen Bioscience (Ann Arbor, MI, USA) using the IncuCyte ZOOM system.

Proliferation: Cells were placed with 30% confluence in a 96-well plate and treated with 50 μg/mL, 5 μg/mL and 0.5 μg/mL ApoE (ab280330, Abcam, UK) in OptiMEM +2% FBS (Thermo Fisher, Germany) or OptiMEM +2% FBS only (control-treatment) for 72 h. The plate was monitored in the IncuCyte ZOOM System (Essen Bioscience, Ann Arbor, MI, USA) with phase contrast and a 2 h imaging pattern. Images were auto-collected and analyzed using the IncuCyte ZOOM software (Essen Bioscience, Ann Arbor, MI, USA) and GraphPad Prism 9.4.1.

Migration: Cells were placed with 100% confluence in a 96-well ImageLockTM plate (Essen Bioscience, Ann Arbor, MI, USA). A wound was created using the Incucyte^®^ 96-Well Woundmaker Tool (Essen Bioscience, Ann Arbor, MI, USA), according to the manufacturer´s instructions. After 2 washes with PBS, the cells were treated with 50 μg/mL, 5 μg/mL and 0.5 μg/mL ApoE (ab280330, Abcam, UK) in OptiMEM +2% FBS (Thermo Fisher, Germany) or OptiMEM +2% FBS only (control-treatment) for 72 h. The plate was monitored in the IncuCyte ZOOM System (Essen Bioscience, Ann Arbor, MI, USA) with phase contrast and a 1 h imaging pattern. Images were auto-collected and analyzed using the IncuCyte ZOOM software (Essen Bioscience, Ann Arbor, MI, USA) and GraphPad Prism 9.4.1.

OLink Protein analysis: For potential proteomic target identification, we used the commercially available OLink^®^ high-throughput, multiplex immunoassay according to the manufacturer’s protocol. Briefly, protein lysates were generated from whole vessel wall samples of 6 non-aneurysmatic popliteal arteries (PA), 16 PAA and 19 AAA samples using Laemmli buffer (normal popliteal arteries were from the same patients as PAAs). Afterwards, 1 μL of undiluted protein was sent to the company (OLink) as requested by the protocol and the cardiovascular panels CVD II and CVD III were used to screen for a total of 180 cardiovascular proteins by Proseek^®^ Multiplex—Proximity Extension Assay. Four proteins served as internal validation control in both panels. Afterward, Proseek generates Cq values, and these were normalized by extension control (dCq_analyte_ = Cq_analyte_ − Cq_Ext Ctrl_), interpolate control (ddCq = dCq_analyte_ − dCq_Interplate Ctrl_) and normalization against a calculated correction factor (NPX = Correction factor − ddCq). Finally, Normalized Protein eXpression (NPX) values are given as Log 2 scale (high NPX value = high protein concentration) as described before [27,28].

Comparative analysis of Affymetrix gene expression and Olink protein quantification data with pathway enrichment analysis: Expression was further analyzed for pathway enrichment through Gene Set Enrichment Analysis (GSEA), using GSEA version 4.3.2 together with KEGG and Molecular Signatures Database Hallmarks gene sets (Appendix A) [29,30,31,32]. A gene-wise ranking score using logFC/abs(logFC) × −log10 (*p*-value) from differential expression results was used to combine significance and direction of change. These scores were used to run GSEA pre-ranked KEGG and Hallmark enrichment analysis with a minimum gene set size of 5. Noteworthy enrichments were considered when categories reached a nominal *p*-value < 0.05 in at least one comparison.

Case report mechanical PAA: Tissue from the “mechanical PAA” was obtained from a 51-year-old male former soldier with a smoking history. Upon occasional swelling and pain in the left knee, especially after exercising, he was diagnosed with an aneurysmatic lesion and an osseous lesion at the back of the tibia directly adjacent to the popliteal artery in MRA and X-ray (Appendix A). A previous trauma or surgery was not reported. The aneurysm was excised and a venous bypass graft was implanted. The osseous lesion was carefully removed. The patient was dismissed after 5 days in good condition and has been symptom-free. The contralateral leg showed no similar pathology.

Statistics: Comparison of Ki67 positive cell turnover was performed by cell counting per high power field and analysis with the Mann–Whitney test. Graphs and figures were created with Excel©, PowerPoint© (Microsoft) (Office Professional Plus 2016) and R (http://www.r-project.org/, accessed on 21 June 2023). Expression values were compared using the Mann–Whitney test with the level of significance at 0.05. Gene Expression Results were analyzed using GraphPad Prism 9.4.1 and student’s *t*-test.

## 3. Results

VSMC phenotype switch and comparison to AAA:

PAA vessel wall samples showed a distinct histomorphology with loss of the typical muscular-type artery architecture including the internal and external elastic lamina in comparison to non-aneurysmatic PA from the same patient (Figure 1A and Appendix A). Ki67 staining reveals a highly significant increased number of proliferating cells within the aneurysmatic vessel wall (Figure 1B). Vimentin and α smooth muscle actin (SMA) positive vascular smooth muscle cells (VSMC) show a loss of desmin as part of the contractile apparatus (Figure 1C and Appendix A). This VSMC phenotype switch from contractile senescent to proliferating dedifferentiated is also a feature of AAA in comparison to non-aneurysmatic aortic wall previously described by others and us [16,33].

Additional features of AAA, such as upregulation of genes in involved in the vascular endothelial growth factor (*VEGF*) receptor and the transforming growth factor ß (*TGF*ß) receptor signaling pathways, are found similarly in PAA as compared to non-aneurysmatic arteries (Figure 1D and Appendix A). Moreover, MMP 9 was found to be significantly upregulated in PAA, whereas pro-apoptotic B-cell lymphoma 2 (*BCL2*) was found upregulated in AAA but unchanged in PAA D and Appendix A). Of note, inflammatory genes tended to be less upregulated in PAA as in AAA (Appendix A).

Immunohistochemistry and expression analysis revealed no such results in a singular case of PAA caused by osseous mechanical strain (case description s. above). Here, the lamellar histomorphology was preserved, desmin was still present in VSMCs and dilated and non-dilated tissue sections were hierarchically clustered with non-aneurysmatic arteries upon expression profiling (Appendix A).

Gene array analysis, expression of apolipoprotein E (ApoE) and pathway enrichment:

An exploratory Affymetrix© array identified genes of the VSMC contractile apparatus down- and MMPs to be dysregulated in PAA compared to PA (Figure 2A and Appendix A, Appendix A). Genes from the apolipoprotein E/C1 locus were also dysregulated. Utilizing qPCR, *APOE* expression in PAA compared to PA was not significant in this small cohort (*p* = 0.1) (Figure 2C). Similar trends were observed for AAA (*p* = 0.11) and PAOD samples (*p* = 0.41) (Figure 2D).

Despite this non-significant gene expression analysis, on the cellular level, APOE co-localized with SMA-positive cells upon immunofluorescence in the media of PAA samples (Figure 2E and Appendix A). Of note, CD-68 positive cells were seen abundantly in PAA samples, too (Appendix A). Genes involved in cholesterol homeostasis, e.g., phospholipid-transporting ATPase (*ABCA1*), were found to be higher expressed in PAA tissue compared PAOD plaque samples (*p* = 0.03) (Appendix A).

Accordingly, a KEGG pathway and hallmarks enrichment analysis of the PAA vs. non-aneurysmatic artery gene expression data identified VSMC contraction (NES −2.2; *p* < 0.05) and TGFß signaling (−2.4; *p* < 0.05) to be negatively correlated with PAA (Figure 2B and Appendix A). VEGF (−1.1; *p* = 0.5) and Apoptosis (ES 1.0; *p* = 0.6) enrichments were not significant.

Additionally, data from a separate proteomic approach (OLink) with PAA, PA and AAA tissue lysates allowed comparative enrichment analysis (Appendix A). Here, comparison of genetic (PAA vs. PA only) and proteomic (PAA vs. PA and PAA vs. AAA) data revealed negative enrichment of APOPTOSIS (Hallmark) at genetic levels, yet positive enrichment at proteomic levels, similar to PAA and AAA (Figure 2F and Appendix A, Appendix A).

APOE decreases cell growth in primary human cell culture by VSMC phenotype restoration:

To test a possible effect of ApoE on VSMCs, a primary human PAA cell culture was established. Here, the expression of VSMC markers, such as SMA (*ACTA2*) or SM22 (*TAGLN*) positivity and a typical SMC shape (indicated by phalloidin staining) was observed in accordance to established and commercially available primary human aortic (hAo)SMCs (Figure 3A and Appendix A). Live cell imaging revealed a dose- and time-dependent inhibitory effect on the proliferation over 72 h in both primary human cell cultures (Figure 3B). Here, 50 μg/mL ApoE significantly decreased the cell proliferation rate (indicated by less confluence) from 48 h onwards compared to untreated cells. No effect on cell migration was observed (Appendix A). Of note, hAoSMCs showed an earlier and more pronounced response than primary human PAA-derived cells.

A VSMC phenotype marker gene expression panel at different time points revealed significant upregulation of myocardin 1 (*MYOCD*) or smoothelin (*SMTN*) upon APOE treatment PAA VSMCs after 48 h (Figure 3C and Appendix A). Again, MYOCD upregulation was seen already at 6 h in hAoSMCs (Figure 3D and Appendix A). Additionally, WB analysis revealed higher abundance of calponin1 in PAA and aortic SMCs after ApoE stimulation for 48 h and similar trends for smoothelin and SM22 (Figure 3E and Appendix A). However, quantitative analysis normalized to ß-actin was not significant.

## 4. Discussion

This study suggests that PAA might have a distinct pathophysiology compared to AAA, PAOD and sole mechanic strain, despite sharing some common pathway involvements. Specifically, APOE overrepresentation in tissues, demonstrated on gene and protein levels for the first time, might be involved in the VSMC phenotype switch and possible restoration of the contractile phenotype (Figure 4). Here, no significant dysregulations in lipid metabolism-associated pathways were identified, emphasizing a potential additional function of APOE for individual patients.

However, disease heterogeneity might be high among PAA patients on histomorphologic and molecular level as reported for AAA patients, possibly in accordance with individual aneurysm diameter [16,34]. Individually, PAA growth and eventual intraluminal thrombus (ILT) formation might be mechanically driven, while others have a distinct pathomechanism [35]. Recently, a microRNA (miRNA) and circulating MMP signature for popliteal involvement have been identified in patients with multiple aneurysm sites [23,36,37,38]. Here, we demonstrate similar involvement of VSMC phenotype switch, VEGF and TGFß signaling and change in histomorphology in PAA as compared to AAA (Figure 1, Figure 2A and Appendix A). Additionally, based on a singular case with possible isolated osseous strain on the popliteal artery, so-called Nora’s lesion, we demonstrate a preserved VSMC phenotype and expression profile in this patient (Appendix A) [10,11,39]. Generally, a modest inflammatory component is described for PAA samples. However, more inflammatory subtypes, as also known for AAA, have been reported [16,23,40,41]. No increased expression of inflammatory genes was observed (Appendix A).

Unique in this study, *APOE* was found more abundant in PAA tissue in comparison to PA and AAA (Figure 2C,D). APOE has been shown to play a detrimental role in atherosclerosis and vascular research—and ApoE^−/−^ mice are the most frequently used pro-atherosclerotic animal model for various vascular diseases [42]. However, depending on the genetic risk alleles for *APOE*, its role differs among patients [43]. Here, we demonstrate a general up(dys-)regulation, yet with mentionable variance (Figure 2A,C). Hence, the individual role in PAA pathogenesis might differ between individuals.

Generally, APOE is involved in cholesterol hemostasis functioning as lipoprotein. Aneurysm and atherosclerotic diseases share many common features, such as matrix remodeling, macrophage homing and calcification. Atherosclerosis to a greater or minor extent is observed in almost every AAA patient. The associated diseases such as coronary artery disease and carotid artery disease, as well as PAOD, are observed more frequently in this cohort [44,45]. However, the onset of disease might differ. Atherosclerosis is considered a disease of the intima, since the early stages of the disease are seen here [46]. AAA is considered a disease of the media based on the early changes observed in animal models [47,48]. The specific role of mechanisms identified in both entities might also differ due to different susceptibility of the vessel wall to the same stimuli such as TGFβ signaling, early in embryogenesis and later in adulthood [49,50]. Here, we show that APOE is more abundant in PAA tissues in comparison to AAA and PAOD (Figure 2B). In addition, genes, known to be upregulated in AAA samples are also found upregulated in PAA and the exploratory proteomic-based pathway enrichment analysis demonstrates similarities of both diseases (Figure 2F and Appendix A). However, further research needs to validate common features and distinct differences between AAA, PAA and atherosclerosis. Moreover, the subcellular mechanistic insights on VSMC phenotype switch of ApoE warrant further elucidation. Most available studies only provide indirect data using apoe^−/−^ mice for mechanistic investigations [51,52].

VSMC phenotype switch is observed in both, aneurysm and atherosclerosis; however, probably to a different extent and with heterogeneous clonality [24,52,53]. Others have reported increased apoptosis by p53 upregulation and loss of VSMCs in PAA samples [54,55]. This aligns with previous reports from AAA, where VSMC loss in matrix remodeling is frequently discussed [23,56]. However, these results must be discussed carefully, since the VSMC phenotype switch might lead to different staining properties. Generally, ECM deposition and vessel wall thickening might be misinterpreted as cellular content loss [47]. VSMC density is still high in PAA samples and cell turnover seems to be high (Figure 1B and Figure 2E). VSMC phenotype switch as a target for aneurysm growth alteration has been successfully demonstrated by us and others [18,57]. Additionally, other target genes, not previously involved in aneurysm research, however, also found to be dysregulated in our samples might be worthwhile investigating in future research, e.g., for their role in specific pathways (Figure 1A and Appendix A, Appendix A).

Limitations of the study are the relatively low numbers of patients and samples included, as well as the only exploratory character of gene expression and proteomic analysis. Hence, conclusions can only be drawn lacking statistical significance and remain speculative. So far, no relevant animal models for PAA have been reported [42]. Regarding the successful implementation of a primary human cell culture in this study, organ-on-a-chip or organoids could be helpful for further research (Figure 3) [58]. However, control cells from human non-dilated “healthy” popliteal artery, currently not available from surgical sites, should be included. Owing to its peripheral localization and the reasonably easy access by endovascular means, local medical, i.e., drug-coated balloon treatment, could be a desirable translational aspect [7,18]. Whether APOE, APOE-modifying chemicals or completely different substances are the most promising targets warrants further research.

## 5. Conclusions

In conclusion, APOE was found to be dysregulated in PAA samples compared to non-aneurysmatic popliteal arteries, AAA and PAOD human tissue samples and co-localized with VSMCs demonstrated to switch their phenotype. APOE stimulation of primary human aortic and popliteal VSMCs reduced cell proliferation and increased contractile cellular marker gene and protein expression. Further research needs to validate these preliminary findings and elucidate future translational applications.

## Figures and Tables

**Figure 1 biomolecules-13-01074-f001:**
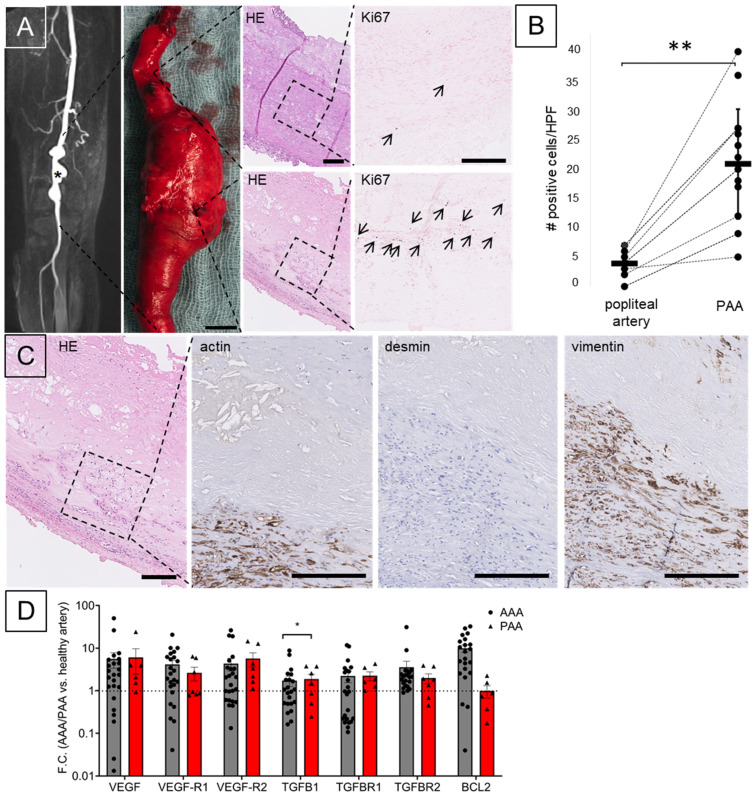
PAA immunohistochemistry, cell proliferation and qPCR results. (**A**) PAA morphology is characterized by vessel dilation (*) on MR-angiography and after surgical resection (scale bar 10 mm). Histological analysis of PAA compared to normal popliteal artery shows calcifications, destruction of elastic fibers, collagen enrichment and angiogenesis on hematoxylin/eosin (HE) staining. (**B**) Ki67 staining (arrow in (A)) and cell counting revealed a significant number of proliferating cells in PAA compared to non-aneurysmatic arteries. (**C**) PAA tissue shows loss of the VSMC contractile marker desmin compared to actin and vimentin (non-aneurysmatic popliteal artery: Appendix A) (scale bar: 100 μm). (**D**) Gene Expression Analysis of PAA (red) and AAA (grey) shown as fold change vs. the respective non-aneurysmatic vessel (AAA vs. aorta: *n* = 24 vs. *n* = 10; PAA vs. popliteal artery: *n* = 7 vs. *n* = 8; PAA vs. AAA: unpaired *t*-test, * *p* < 0.05, ** *p* < 0.01).

**Figure 2 biomolecules-13-01074-f002:**
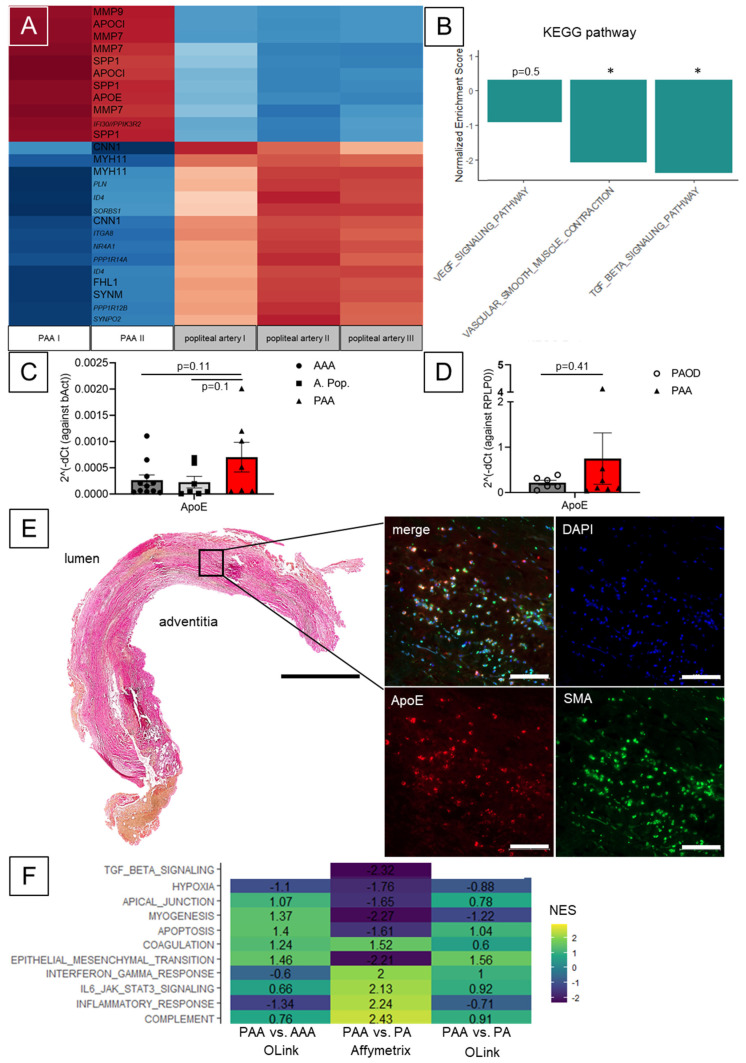
Affymetrix Analysis, Pathway enrichment, comparative ApoE expression and PAA immunofluorescence. (**A**) Gene expression heatmap (top 25 hits, cutout from Appendix A) with hierarchical clustering for PAA vs. popliteal artery based on lowest adjusted *p*-value (all > 0.05: Appendix A) (annotations: Appendix A). (**B**) Selected examples of KEGG pathway enrichment based on Affymetrix data (NES: normalized score shown in (F), * = *p* < 0.05). (**C**) RT-qPCR results of ApoE expression in PAA (red) compared to popliteal artery (light grey), AAA (dark grey) and (**D**) PAOD (grey) shown as 2^dCt values against the respective housekeeping gene (*n* = 11 AAA, *n* = 7 A. pop., *n* = 7 PAA; *n* = 6 PAOD, *n* = 7 PAA; mean +/− SEM; unpaired *t*-test, * *p* < 0.05). (**E**) PAA tissue sample (HE; scale bar 2 mm) and immunofluorescence with APOE (red) and SMA (green)-positive cells co-localization (scale bar 100 μm). (**F**) Exemplary comparative hallmarks pathway enrichment analysis of Affymetrix gene expression data and OLink Protein expression data for PAA vs. popliteal artery (PA) and PAA vs. AAA (all *p*-value < 0.0.5) (complete enrichment sets shown in Appendix A).

**Figure 3 biomolecules-13-01074-f003:**
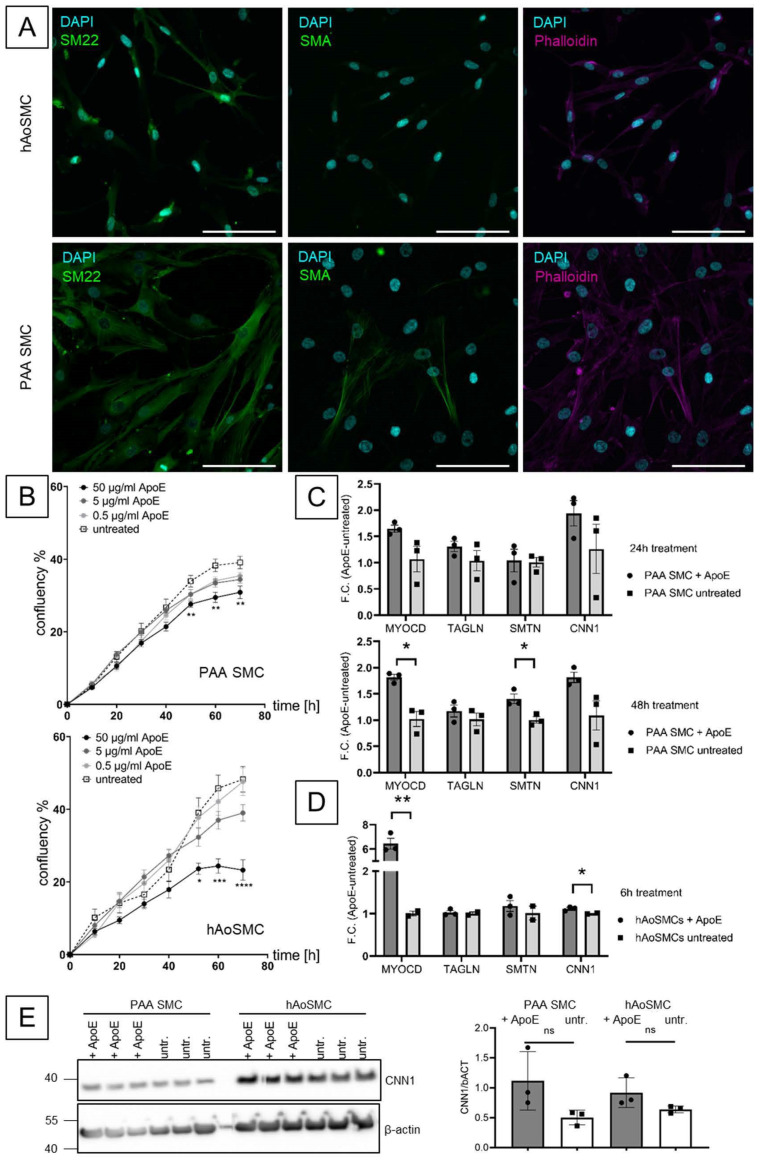
PAA primary cell culture immunofluorescence, Incucyte© live cell imaging and contractile VSMC marker genes analysis. (**A**) PAA cells show similar morphology and expression of SMC-specific markers (SM22, SMA) and overall morphology (phalloidin) when compared to AoSMCs (scale bar 100 μm). (**B**) Live cell imaging depicts changes in confluency over time with different ApoE treatment concentrations for human aortic smooth muscle cells (left) and primary PAA patient-derived cells (two-way ANOVA treated vs. untreated, * *p* < 0.05, ** *p* < 0.01, *** *p* < 0.001, **** *p* < 0.0001; Mean +/− SEM). (**C**) Exemplary RT-qPCR results of VSMC contractile marker gene expression at 24 h and 48 h after ApoE treatment (dark grey vs. no treatment light grey) for PAA primary cells and (**D**) at 6 h for human aortic SMCs (both 50 μg/mL ApoE) (unpaired *t*-test) (full panel in Appendix A). (**E**) WB and quantification for CNN1 normalized to ß-actin for PAA SMCs and hAoSMCs upon ApoE stimulation.

**Figure 4 biomolecules-13-01074-f004:**
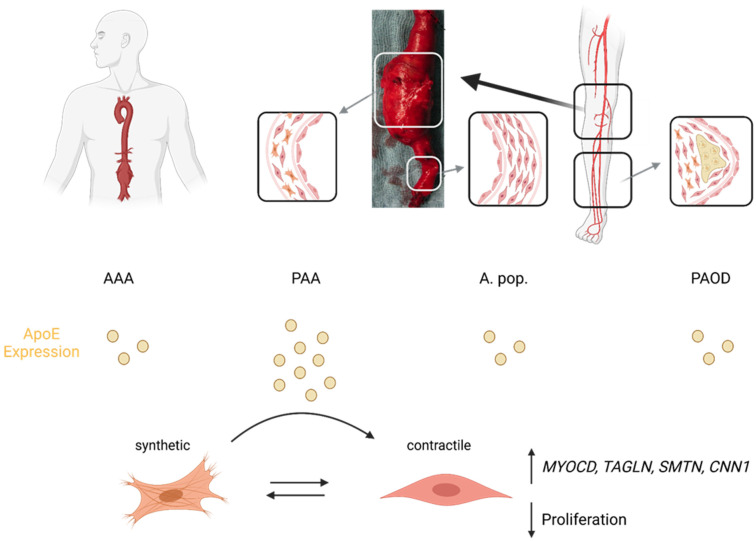
Illustrated abstract. Schematic depiction of proposed speculative mechanism (created with BioRender.com). ApoE is more abundant in tissues from PAA patients compared to non-dilated A. pop. or other vascular diseases (AAA and PAOD). Treatment of VSMCs with ApoE leads to decreased proliferation and increased expression of contractile markers such as MYOCD, TAGLN, SMTN and CNN1on RNA and protein levels. Overall, this study suggest that ApoE might be involved in contractile phenotype restoration of popliteal VSMCs and thus counteract aneurysm formation (in individual patients).

## Data Availability

Not applicable.

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
