# Peer review of "Apolipoprotein E (ApoE) Rescues the Contractile Smooth Muscle Cell Phenotype in Popliteal Artery Aneurysm Disease"

_biomolecules, 2023, doi:10.3390/biom13071074_

Round 1

Reviewer 1 Report

Dear authors,

congratulations on an exciting and novel manuscript! One minor drawback of the manuscripts inherits the low number of samples, this is however explained by the entity of the analysis.

No further remarks.

Advise accept the manuscript.

none; 

Author Response

REVIEWER 1

Dear authors,

congratulations on an exciting and novel manuscript! One minor drawback of the manuscripts inherits the low number of samples, this is however explained by the entity of the analysis.

No further remarks.

Advise accept the manuscript.

Thank you very much for the thorough assessment and friendly review of our manuscript.

Reviewer 2 Report

Pouli et al, performed an exploratory study aiming the elucidation of the molecular mechanisms that underlying the progression of popliteal artery aneurysm using a multitude of methods, including Olink proteomics, immunohistochemistry, mRNA expression profiling and qPCR. Although the study is of interest to the field and deserves some credit some serious issues should be addressed:

Despite the large number of different analytical methods applied a common thread through all the data is that very few significant differences in tissue types (PAA, AAA etc.) have been found, indicating that 1) the data have too much variance or 2) too few samples are included in the experiments, which makes it difficult to draw precise conclusions. Therefore, the study should be considered as a preliminary one.

Specific comments:

 Some of the figures and corresponding legends are very confusing and not very informative, e.g. Fig. 2A-D, 2F. Moreover, it´s not clear from this figure if APOE is significantly regulated or not.

 As consequence, the proposed mechanism depicted in figure 4 is too speculative and not really experimentally supported.

Needs some improvement

Author Response

REVIEWER 2

Comments and Suggestions for Authors

Pouli et al, performed an exploratory study aiming the elucidation of the molecular mechanisms that underlying the progression of popliteal artery aneurysm using a multitude of methods, including Olink proteomics, immunohistochemistry, mRNA expression profiling and qPCR. Although the study is of interest to the field and deserves some credit some serious issues should be addressed:

Thank you very much for the thorough assessment of our manuscript. Please find below a detailed point-to-point response to all your questions raised. We have included modifications in the revised version of the manuscript, which has gained visibility and clarity through the review process.

Despite the large number of different analytical methods applied a common thread through all the data is that very few significant differences in tissue types (PAA, AAA etc.) have been found, indicating that 1) the data have too much variance or 2) too few samples are included in the experiments, which makes it difficult to draw precise conclusions. Therefore, the study should be considered as a preliminary one.

Thank you very much for this comment. We are very aware of the limitations of this manuscript and especially the methods applied regarding the low sample size. However, popliteal aneurysm is a rare entity and especially resection of the aneurysm and adjunct (healthy) non-dilated popliteal artery is performed infrequently. So far nothing is known about specific pathomechanisms involved in PAA pathogenesis.

We do consider our manuscript of preliminary nature, due to these limitations and thus conclusions should only be drawn very carefully. We have addressed this issue again in the revised version by

  • Modification of the abstract and the conclusion section emphasizing the preliminary nature of the manuscript.

Specific comments:

Some of the figures and corresponding legends are very confusing and not very informative, e.g. Fig. 2A-D, 2F. Moreover, it´s not clear from this figure if APOE is significantly regulated or not.

We apologize for the confusion. As mentioned above, the low sample renders statistical analysis difficult to interpret. We thus aimed to show various associations on different levels to account for this shortcoming.

However for the revised version of the manuscript, we have

  • included p-values in gene expressions results in Fig. 2C,D
  • modified the figure legend for better explanation regarding the statistical significance of the visualized data.

As consequence, the proposed mechanism depicted in figure 4 is too speculative and not really experimentally supported.

We agree that the mechanism is speculative – isn’t this part of the interpretation of this preliminary data?

However, we were able to experimentally show an effect of ApoE on VSMCs regarding proliferation and phenotype switch with different means (gene expression/WB), based on the non-significantly demonstrated assumption that ApoE might be available at higher abundance in PAA patients compared to other vascular diseases.

In the revised version of the manuscript

  • we have modified the title and added a specific figure legend to Fig. 4 for clarification.

Reviewer 3 Report

The manuscript by Pauli et al. aimed to investigate the effect of apoE on the phenotype of contractile smooth muscle cells in popliteal artery aneurysm disease. This manuscript is interesting; however, there are several deficits in the current form of this manuscript needing tom be experimentally addressed.

1. The effect of apoE on the VSMC proliferation and migration needs to be experimentally addressed.

2. VSMC contractile markers should be examined by western blot analysis.

3. Were the levels of apoE changed in popliteal artery aneurysm disease as compared to that of normal artery?

4. What is the mechanism underlying the apoE-mediated effect on the phenotype switch of VSMA? Authors should, at least, discuss this point in revised manuscript.

5. In figure 2, the levels of many molecules were changed in clinical samples from patients with popliteal artery aneurysm disease, authors should , at least, discuss their possible contribution in the pathogenesis of popliteal artery aneurysm disease.

Author Response

REVIEWER 3

Comments and Suggestions for Authors

The manuscript by Pauli et al. aimed to investigate the effect of apoE on the phenotype of contractile smooth muscle cells in popliteal artery aneurysm disease. This manuscript is interesting; however, there are several deficits in the current form of this manuscript needing tom be experimentally addressed.

Thank you very much for the review of our manuscript and the suggestions in order to improve its overall quality. Please find below a detailed point-to-point reply to all issues raised including substantial modifications to the revised version of the manuscript as well as new data.  

The effect of apoE on the VSMC proliferation and migration needs to be experimentally addressed.

Thanks for the insightful comment. In our initial study, we have only looked at cell proliferation, since one initial observation was the high abundance of Ki67 positive cells in PAA samples (Fig. 1A). Here, we observed a reduced confluence rate of both, popliteal aneurysm and aortic VSMCs upon treatment with ApoE (Fig. 3B).

For the revised version of the manuscript we have performed an additional migration assay for both VSMC entities using the same ApoE concentrations applied before and do not see any effect on cell migration.

  • the data was added to Supplement Figure 4 (SF 4D)
  • the methodology was added to the M&M section
  • the results section was modified accordingly

VSMC contractile markers should be examined by western blot analysis.

Again, thank you for the comment. We examined the protein abundance of SMTN, CNN1 and SM22, in accordance with the significantly regulated genes by Western Blot (WB) for the revised version of the manuscript and strongly feel that the overall visibility of the manuscript has gained tremendously.

We did not perform WB of MYOCD as this is a transcription factor and not a contractile protein and leads to the transcription of contractile marker mRNA, which is then later translated into proteins. Also the expression of transcriptions factors might be timely very limited. We decided on one time-point (48h) in order to meet the journal’s timeline for revision – and therefore went with stable contractile proteins SMTN, CNN1 and SM22.

For the revised version of the manuscript we have performed an additional WB for both VSMC entities

  • included WB data in Fig. 3 and Suppl. Fig. 3 for SMTN, CNN1 and SM22.
  • included the methodology applied in the M&M section.
  • modified the results section accordingly.
  • included the complete membranes as an additional Suppl. Fig. 5.
  • modified the conclusion accordingly.

Were the levels of apoE changed in popliteal artery aneurysm disease as compared to that of normal artery?

Please see also the according comment from Reviewer 2.

As mentioned above, the low sample renders statistical analysis difficult to interpret. We thus aimed to show various associations on different levels to account for this shortcoming. Generally, the gene expression analysis did not show significance on the individual or the affymetrix expression analysis, yet, ApoE was chosen as target of interest due its surprising hit amongst the most dysregulated targets (Fig 2A). As included in the discussion section, the levels of ApoE show a high variance in PAA tissue samples, possibly reflecting individual disease burden or entities among patients.

However, for the revised version of the manuscript, we have

  • included p-values in gene expression results in Fig. 2C,D
  • modified the figure legend for a better explanation regarding the statistical significance of the visualized data.
  • added to the discussion section

What is the mechanism underlying the apoE-mediated effect on the phenotype switch of VSMA? Authors should, at least, discuss this point in revised manuscript

This is an interesting question. We have not investigated subcellular mechanistic links of ApoE in this study. Screening through the literature, no studies report direct stimulation of VSMC with ApoE and the effect on phenotype switch. However, a variety of studies report different effects of various genes, transcription factors, and drugs on apoe -/- mice and their respective VSMCs. However, we do not consider this adequate to suggest any subcellular mechanism for the results presented here.

For the revised manuscript, we have

  • added to the discussion section for this specific point, also adding additional references.

In figure 2, the levels of many molecules were changed in clinical samples from patients with popliteal artery aneurysm disease, authors should , at least, discuss their possible contribution in the pathogenesis of popliteal artery aneurysm disease.

We have chosen to investigate APOE in more detail after the initial gene profiling due to the surprising observation of this specific genetic locus including APOE/APOCI popped up in our analysis and has not been investigated in the context of aneurysm research before.

We agree that the other genes included in Fig. 2A are also worth being mentioned in the discussion. All the gene names are listed in Suppl. Table II.

For the revised version of the manuscript we have

  • added a specific sentence to the discussion section.

Reviewer 4 Report

Major Comments

1.    In in vitro studies, isolated SMCs were used.  Primary PAA SMCs were isolated by the authors and cultured in complete DMEM/F12 medium while hAoSMCs were bought commercially and cultured in Smooth Muscle Cell Growth Medium.  The differences reported could be due to the differences in culturing conditions.  In addition, control cells must be made in the same way as PAA SMCs from normal popliteal artery, especially because the study seems to indicate that SMCs from aorta and popliteal artery respond differently to experiments.

2.    If statistical analyses show no difference, the two data sets are not different.  However, in many places, the authors suggest there were differences.  For example, the authors state, “Genes from the apolipoprotein E/C1 locus were also upregulated. Utilizing qPCR, APOE was found to be upregulated non-significantly in PAA compared to PA (p=0.1) (Fig. 2C). Similar trends were observed for AAA (p=0.11) and PAOD samples (p=0.41)” What does “upregulated non-significantly” mean?

3.  The coinclusions are suggestive, but not strongly convincing.

Minor comments.

1.     Throughout the text, insert a space between the unit (such as mg, ml, etc.) and number.

2.    AAA is defined as both “abdominal aortic aneurysm” and “ascending and infrarenal aorta”  Use different abbreviations.

3.    “HPF:40x magnification”   Is 40x the total magnification?  Or do you mean using a 40x objective lens?  Also, specify the cameras used for image recording.

4.    For RNA isolation, the authors state, “Tissues were cut in ~50mg pieces.”  Was each piece actually weighed?  Give dimensions rather than weight.

5.    Gene expression was normalized to Rplp0.  Show that this control gene expression level is the same among various samples.  Differences in cell proliferation could change the level of this gene expression.

6.    Line 215.  The text reads, “…undiluted protein was sent to the manufacturer.”  What do you mean by “manufacturer”?

7.    Figure 1D.  Visually, it is difficult to see difference in TGFB1 expressions while BCL2 expressions are statistically the same.

8.    Fig. 3C, D.  Again, visually, it is difficult to believe no statistical differences in CNN1 after 24 and 48 hours while there is at 6 hours.

9.    Figure 4 must have a more detailed legend, especially tieing the ApoE expression with SMC phenotypes.  

Some typos and grammatical errors.  Needs English editing.

Author Response

Please find a detailed point-to-point response in the word file attached. 

Round 2

Reviewer 2 Report

No further comments to the authors

Reviewer 3 Report

Authors have addressed my comments.

Reviewer 4 Report

Authors responded to my comments adequately.

There are still language issues.  The journal editor needs to work hard at it.

Line 197, "Tissues were cut in...... on dry ice using a small dimension scale."  What is a small dimension scale that was used cut the tissue? 

Please see above